# EFFICIENTLY DEPLOYING LLMS WITH CONTROLLED RISK

## ABSTRACT

Deploying large language models in production requires simultaneous attention to efficiency and risk control. Prior work has shown the possibility to cut costs while maintaining similar accuracy, but has neglected to focus on risk control. By contrast, here we present hierarchical chains with multi-level abstention (HCMA), which use model-intrinsic uncertainty to delegate queries along the LLM intelligence hierarchy, enabling training-free model switching based solely on black-box API calls. Our framework presents novel trade-offs between efficiency and risk. For example, deploying HCMA on MMLU cuts the error rate of Llama3 405B by 30% when the model is allowed to abstain on 20% of the queries. To calibrate HCMA for optimal performance, our approach uses data-efficient logistic regressions (based on a simple nonlinear feature transformation), which require only 50 or 100 labeled examples to achieve excellent calibration error (ECE), cutting ECE by 50% compared to naive Platt scaling. On free-form generation tasks, we find that chain-of-thought is ineffectual for selective prediction, whereas zero-shot prompting yields drives error to 0% on TruthfulQA at high abstention rates. As LLMs are increasingly deployed across computing environments with different capabilities (such as mobile, laptop, and cloud), our framework paves the way towards maintaining deployment efficiency while putting in place sharp risk controls.

## 1 INTRODUCTION

Given the recent excitement around large language models, researchers have been intensely focused on evaluating potentially transformative applications across domains. However, as the technology matures and its scale increases, *efficiency* starts to matter. In addition, real-world consequences begin to emerge, calling for greater *risk control*.

The NLP community has addressed the need for greater efficiency by optimizing the internal mechanics of the transformer-based neural networks (Dao et al. (2022)), developing new sampling schemes (Leviathan et al. (2023)), performing model compression and distillation (Hinton et al. (2015), Xu et al. (2024)), caching model responses (Bang (2023)), and routing queries between different models (Chen et al. (2023b), Jiang et al. (2023), Hu et al. (2024), Aggarwal et al. (2024), Ding et al. (2024)). On risk control, researchers have considered probabilistic calibration and various approaches to *selective* prediction, in which a model is allowed to abstain from queries based on a confidence signal such as single-token or sequence-level probabilities (Xin et al. (2021)), entropy-based scores and consistency measures (Manakul et al. (2023), Kuhn et al. (2023), Chen et al. (2024)), lightweight probes on hidden layer embeddings (Azaria & Mitchell (2023), Kossen et al. (2024)), or outputs of separate neural networks trained to predict correctness (Xin et al. (2021), Yoshikawa & Okazaki (2023), Varshney & Baral (2023), Gupta et al. (2024), Cobbe et al. (2021), Kadavath et al. (2022)).

However, previous work has not simultaneously addressed efficiency and risk control. Following previous routing approaches in which small models pass queries to larger models based on a confidence threshold (Kag et al. (2023), Chen et al. (2023b), Aggarwal et al. (2024), Wang et al. (2024), Ding et al. (2024), Sakota et al. (2024)), we incorporate selective prediction by adding "abstention thresholds" that let each model reject queries on behalf of the whole LLM system. In this paper, we explore the use of LLM token probabilities to instantiate this system, which we call "hierarchical chains with multi-level abstention." Although the most performant LLM confidence signals – when considered in isolation – are based on hidden layer embeddings, repeated sampling, and neural-network correctness predictors, these approaches are not suitable for meeting our twin goals of 1) improving efficiency while simultaneously imposing risk control, and 2) creating a framework based on black-box API calls that is easily adaptable to new tasks without requiring much data. By contrast, model-intrinsic probabilities have the advantage that they do not require white-box access to model internals, are widely available as part of API calls to third-party LLM inference providers, do not require large amounts of training data, and avoid compute overhead from repeated sampling.

## 2    RELATED WORK

- **Uncertainty Quantification**: our framework of hierarchical LLM chains relies on uncertainty quantification to distinguish easy queries from difficult ones. Previous work has shown that on text classification tasks (such as MMLU), the maximum softmax probability effectively predicts mistakes, even if it is not calibrated (Hendrycks & Gimpel (2018), Plaut et al. (2024)). On natural language generation, the community has made progress with several different approaches including prompting an LLM to verify whether a response to a query is correct (Lin et al. (2022), Kadavath et al. (2022), Xiong et al. (2024)), assessing the consistency between multiple sampled LLM outputs (Lin et al. (2024), Manakul et al. (2023), Kuhn et al. (2023), Aichberger et al. (2024), Nikitin et al. (2024)), and training lightweight probes on models' internal states (Ren et al. (2023), Azaria & Mitchell (2023), Chen et al. (2024), Kossen et al. (2024)).

- **Probability Calibration**: several methods have been proposed to make probabilistic predictions more calibrated (such that a forecast with probability 0.8 comes true about 80% of the time), including Platt scaling (Platt (1999)), temperature scaling (Guo et al. (2017)), isotonic regression (Zadrozny & Elkan (2002)), Bayesian binning (Naeini et al. (2015)), and more. Recent work has shown that state-of-the-art large language models tend to have poor calibration, especially after reinforcement learning from human feedback (RLHF) (Ouyang et al. (2022), OpenAI et al. (2024), Plaut et al. (2024)). Due to its simplicity, temperature scaling has emerged as the most popular post-processing method for calibration, though for the purposes of rigorous risk control it suffers from a lack of statistical grounding.

- **Selective Prediction**: this field studies the performance of machine learning models when they are allowed to abstain on difficult queries. With the machine learning community, foundational work of El-Yaniv & Wiener (2010), Geifman & El-Yaniv (2017), and others was centered on image classification. In NLP, selective prediction has originally attracted limited interest (Xin et al. (2021), Varshney et al. (2022), Varshney & Baral (2023), Ren et al. (2023), Yoshikawa & Okazaki (2023), Chen et al. (2023a)) but today has become largely subsumed in the emerging field of uncertainty quantification for LLMs, which uses similar performance metrics. We wish to highlight the SGR algorithm (Geifman & El-Yaniv (2017)) as a compelling technique for endowing selective prediction methods (including ours) with provable risk guarantees.

- **Routing between Language Models**: recent literature has reported cost savings (and, sometimes, performance gains) from routing queries between different language models. In this area, many approaches focus on *horizontal* routing between a collection of small language models in the hope of leveraging complementary strengths (Lu et al. (2023), Hari & Thomson (2023), Lee et al. (2024), Sakota et al. (2024), Wang et al. (2024)). Other works consider *vertical* routing in which queries are explicitly first sent to a smaller model, which forwards difficult queries to a larger model (Wang et al. (2023), Kag et al. (2023), Chen et al. (2023b), Yue et al. (2024), Aggarwal et al. (2024), Ding et al. (2024)). However, these approaches have focused on lowering the cost of LLM inference without simultaneously tackling selective prediction.

## 3    OUR CONTRIBUTIONS

First, we provide a novel formula explaining why uncertainty-based delegation between language models works. Second, we introduce a nonlinear feature transformation that makes Platt scaling a highly effective calibration technique for LLM token probabilities, providing an alternative to temperature scaling grounded in a rigorous statistical model (logistic regression). Third, we present hierarchical chains with multi-level abstention (HCMA), the first (to our knowledge) routing LLM model that incorporates selective prediction alongside cost efficiency. By computing the full two-dimensional Pareto frontier of efficient HCMA configurations on MMLU, using formulas we derive, we show that an LLM routing approach based on model-intrinsic probabilities improves over the selective prediction performance of single LLMs, in terms of strengthening risk control without paying more, or achieving similar risk control at the same (or lower) price. Finally, we note that chain-of-thought prompting yields poor selective prediction performance on TruthfulQA compared to zero-shot prompting, calling for caution when applying established prompting techniques in LLM uncertainty quantification.

## 4    MATHEMATICAL FRAMEWORK

We present an LLM system called "hierarchical chains with multi-level abstention," which is based on delegating queries from small to large models, but allowing each model to abstain on behalf of the entire chain. First, we examine the act of delegation and explain why it works.

### 4.1 WHY DOES DELEGATION WORK?

By delegation, we mean passing queries from a small model to a larger model based on estimated difficulty. Prior work has shown that this strategy outperforms random query assignment, but has not provided an explanation (Xin et al. (2021)). Although the result is intuitively plausible, the situation is nuanced because differently sized models seem to share a common sense of difficulty. Figure 1 shows this trend by exhibiting logistic regressions that predict correctness of Llama3 8B, 70B, and 405B on MMLU **solely based on the transformed probability of the 8B model**.

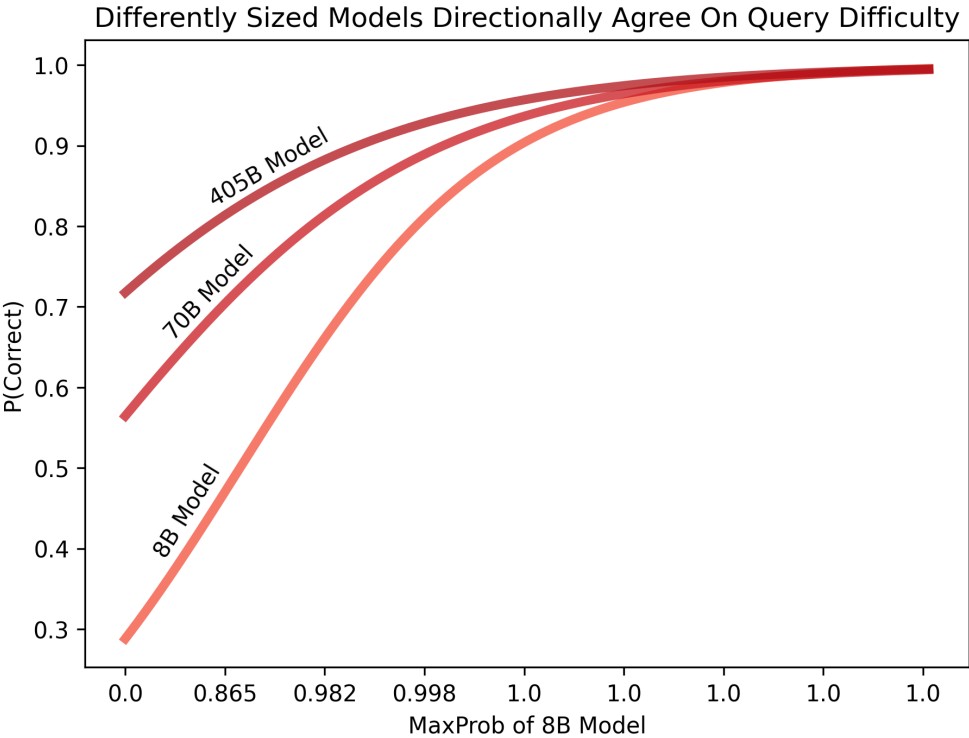

Figure 1: Model-intrinsic uncertainty aligns across Llama3 8B, 70B, and 405B models on MMLU, seemingly reflecting a shared sense of difficulty, but with the larger models more resistant to it. The figure shows logistic regressions fit to the transformed probabilities of the 8B model.

Given that differently sized models share a common notion of difficulty, the effectiveness of delegation depends on the net result of two opposing forces: the small model makes *fewer mistakes* because it only answers the easy queries, but the large model makes *more mistakes* since it answers only difficult queries. Thus, the effectiveness of delegation rests on the large model being *less sensitive* to incremental difficulty than the small model.

The proposition below frames our reasoning in mathematical terms:

**Proposition 1.** *Compared to randomly assigning queries to a small language model $\mathcal{M}_{sm}$ and $\mathcal{M}_{lg}$, the reduction in error from forwarding queries based on a delegation decision $D$ is*

$$\Delta E = Cov(\mathbf{1}_D, \mathbf{1}_{\mathcal{M}_{bg} \text{ makes an error}}) - Cov(\mathbf{1}_D, \mathbf{1}_{\mathcal{M}_{sm} \text{ makes an error}}). \tag{1}$$

In this formula, the covariances $\text{Cov}(\mathbf{1}_D, \mathbf{1}_{\mathcal{M}_{sm} \text{ makes an error}})$ and $\text{Cov}(\mathbf{1}_D, \mathbf{1}_{\mathcal{M}_{bg} \text{ makes an error}})$ measure the impact of the delegation decision on each model's propensity to make a mistake. Empirically, both covariances are positive if $D$ is based on query difficulty, since both models' performance degrades with incremental difficulty. However, as long as $\text{Cov}(\mathbf{1}_D, \mathbf{1}_{\mathcal{M}_{sm} \text{ makes an error}}) > \text{Cov}(\mathbf{1}_D, \mathbf{1}_{\mathcal{M}_{bg} \text{ makes an error}})$, meaning that the smaller model is more sensitive to difficulty, the change in error in (1) is negative, and delegation outperforms random query assignment.

### 4.2 HIERARCHICAL CHAINS WITH MULTI-LEVEL ABSTENTION

A HCMA consists of a chain $\mathcal{M}_1 \to ... \to \mathcal{M}_k$ of LLMs. A query is first sent to $\mathcal{M}_1$, which makes a decision whether to provide an answer to the query (ACCEPT), delegate it to a larger model (DELEGATE), or reject the query outright

(REJECT). This decision is based on an estimated probability $\hat{p}_{\mathcal{M}_1}(x)$ of correctness on query $x$. Models $\mathcal{M}_2, ..., \mathcal{M}_k$ behave analogously. Importantly, when a model rejects a query, the rejection applies to the whole HCMA.

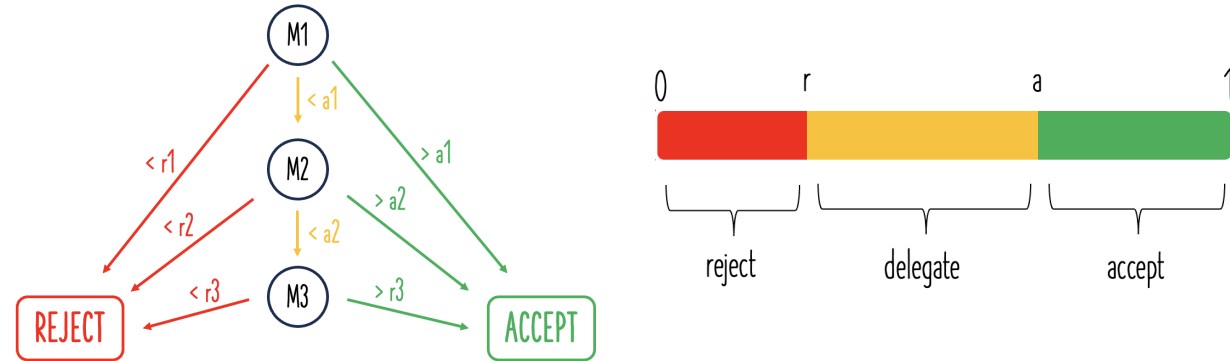

Figure 2: Illustration of a hierarchical chain with multi-level abstention (left), which consists of three models M1-M3, and the action space of an individual model in the chain (right), as a function of the estimated correctness probability.

Each model $\mathcal{M}_j$ ($j < k$) makes its decision based on the estimated correctness probability and two tunable thresholds $r_j$ and $a_j$. The model rejects the query if estimated correctness probability is below $r_j$, delegates the query if it is between $r_j$ and $a_j$, and accepts the query if the estimated correctness probability exceeds $a_j$. This strategy results in the model-specific policies

$$\pi_{\mathcal{M}_j}(\hat{p}_{\mathcal{M}_j}) = \begin{cases} \text{REJECT} & \text{if } \hat{p}_{\mathcal{M}_j} < r_j, \\ \text{DELEGATE} & \text{if } \hat{p}_{\mathcal{M}_j} \geq r_j \text{ but } \hat{p}_{\mathcal{M}_j} < a_j, \\ \text{ACCEPT} & \text{if } \hat{p}_j \geq a_j, \end{cases} \quad (2)$$

for models $j < k$. The last model $\mathcal{M}_k$ cannot delegate to a larger model, so it only rejects or accepts queries based on a single threshold $r_k$. All in all, an HCMA with $k$ models has $2k - 1$ tunable configuration parameters. Figure 2 illustrates the structure of an HCMA and the action space of an individual model.

### 4.3 ANALYZING SYSTEM PERFORMANCE: EFFICIENCY AND RISK

To measure performance of an HCMA in terms of efficiency and risk control, we propose the following metrics: error rate, cost, and *abstention rate* (the fraction of queries on which the chain refuses to give an answer, in order to control risk). We measure cost in terms of either money ($) or latency (ms). Since a HCMA delegates queries sequentially along the chain, costs add up as a query propagates more deeply into the chain. For example, suppose the HCMA $\mathcal{M}_1 \rightarrow \mathcal{M}_2 \rightarrow \mathcal{M}_3$ has model-specific costs $c_1$, $c_2$, and $c_3$. Then, whenever the query ends at model $i$, the effective cost is $C_i = \sum_{j=0}^{i} c_j$.

The following proposition provides formulas for the HCMA performance metrics, based on the graph structure in Figure 2.

**Proposition 2.** *Consider an HCMA $\mathcal{M}_1 \rightarrow \mathcal{M}_2 \rightarrow ... \rightarrow \mathcal{M}_k$ with parameters and $\{r_j\}_{j=1}^{k}$, $\{a_j\}_{j=1}^{k-1}$ and model-specific costs $c_1, ..., c_k$. Based on the true joint distribution $p(x, y)$ of queries $x$ and ground truth responses $y$, the population performance metrics are*

$$\mathbb{P}(\text{Error}) = \sum_{j=1}^{k} \mathbb{P}_{(x,y) \sim p(x,y)}(\pi_1 = \text{DELEGATE}, ..., \pi_{j-1} = \text{DELEGATE}, \pi_j = \text{ACCEPT}, Y_j \neq y), \quad (3)$$

$$\mathbb{P}(\text{Abstain}) = \sum_{j=1}^{k} \mathbb{P}_{(x,y) \sim p(x,y)}(\pi_1 = \text{DELEGATE}, ..., \pi_{j-1} = \text{DELEGATE}, \pi_j = \text{REJECT}), \quad (4)$$

$$\mathbb{E}[\text{Cost}] = \sum_{j=1}^{k} \mathbb{P}_{(x,y) \sim p(x,y)}(\pi_1 = \text{DELEGATE}, ..., \pi_{j-1} = \text{DELEGATE}, \pi_j \neq \text{DELEGATE}) C_j, \quad (5)$$

*where $Y_j$ is the response of the $j$-th model and $y$ the assumed ground truth on query $x$. The constants $C_j = \sum_{\xi=1}^{j} c_\xi$ are the effective costs when model $\mathcal{M}_j$ resolves a query.*

To evaluate HCMA performance in practice, we estimate the chain's performance metrics using Monte Carlo approximations as

$$\widehat{\mathbb{P}(\text{Error})} = \sum_{j=1}^{k} \sum_{i=1}^{n} \frac{1}{n} \, \mathbb{I}[\hat{p}_1(x^{(i)}) \in [r_1, a_1), ..., \hat{p}_{j-1}(x^{(i)}) \in [r_{j-1}, a_{j-1}), \hat{p}_j(x^{(i)}) \geq a_j] \, (1 - \hat{p}_j(x^{(i)})), \quad (6)$$

$$\widehat{\mathbb{P}(\text{Abstain})} = \sum_{j=1}^{k} \sum_{i=1}^{n} \frac{1}{n} \, \mathbb{I}[\hat{p}_1(x^{(i)}) \in [r_1, a_1), ..., \hat{p}_{j-1}(x^{(i)}) \in [r_{j-1}, a_{j-1}), \hat{p}_j(x^{(i)}) < r_j], \quad (7)$$

$$\widehat{\mathbb{E}[\text{Cost}]} = \sum_{j=1}^{k} \left( \sum_{i=1}^{n} \frac{1}{n} \, \mathbb{I}[\hat{p}_1(x^{(i)}) \in [r_1, a_1), ..., \hat{p}_{j-1}(x^{(i)}) \in [r_{j-1}, a_{j-1}), \hat{p}_j(x^{(i)}) \notin [r_j, a_j)] \right) C_j, \quad (8)$$

where the $\mathbb{I}[\cdot]$ are indicator variables and the $\hat{p}_j$ are the fitted correctness predictors (based on a small training set, independent from the data for which the performance estimates are computed). Note that in the formulas above, we write $a_k = r_k$ interchangeably for the rejection threshold of the last model, to maintain consistency and avoid a special case.

Importantly, the joint probabilities in (4)-(6) and (7)-(9) do not approximately factorize into independent terms, since the estimated correctness probabilities across models are highly correlated (see Figure 1).

### 4.4 Making Model-Intrinsic Uncertainty Work For HCMAs

The policy (2) of each model $\mathcal{M}_j$ in a HCMA $\mathcal{M}_1 \rightarrow ... \rightarrow \mathcal{M}_k$ requires an estimated probability of correctness $\hat{p}_{\mathcal{M}_j}(\cdot)$ for $j = 1, 2, ..., k$. As we rely on each model's intrinsic token probabilities $p_{\text{raw}}$ for uncertainty quantification, obtaining $\hat{p}$ boils down to calibrating $p_{\text{raw}}$.

Different calibration techniques have been employed for post-processing the conditional probabilities of large language models. Originally proposed for computer vision networks, temperature scaling (Guo et al. (2017)) has emerged as a favored method because of its simplicity and high performance (Jiang et al. (2021)). However, Platt scaling (Platt (1999)) is preferable for rigorous risk control, since it is based on a statistical model – logistic regression – for which statisticians hafve developed an established catalog of confidence intervals and model diagnostics. Unfortunately, standard Platt scaling does not work well for LLMs because its conditional probabilities tend to form a tight cluster near 1.0, reflecting overconfidence (OpenAI et al. (2024), Plaut et al. (2024)).

However, it is possible to make Platt scaling work very well by spreading out the clusters of overconfident probabilities. We propose simple nonlinear transformations that accomplish this by introducing asymptotes near $p_{\text{raw}} = 0$ and $p_{\text{raw}} = 1$. Specifically, on multiple-choice QA, we transform the maximum softmax probability by

$$p_{\text{tr}}(p_{\text{raw}}) = \log\left(\frac{1}{1 - p_{\text{raw}}}\right), \quad (9)$$

to obtain transformed probabilities $p_{\text{tr}}$.

On open-ended QA, we reduce the correctness prediction problem to binary classification by adopting the P(True) strategy proposed by Kadavath et al. (2022). In this approach, after calling a model $\mathcal{M}$ to answer the query, we call $\mathcal{M}$ again with a verification prompt to output "Y" or "N" based on whether its originally generated answer is correct. Since this is a binary classification with the constraint $p_{\text{raw}}(\text{"Y"}) = 1 - p_{\text{raw}}(\text{"N"})$, we consider a different transformation that spreads overconfident probabilities for "N" and "Y" into the negative and positive reals. In terms of $p = p_{\text{raw}}(\text{"Y"})$, we apply the transformation

$$p_{\text{tr}}(p) = \begin{cases} \log(\frac{1}{1-p}) & \text{if } p \geq 0.5 \\ \log(2) - \log(\frac{1}{p}) & \text{if } p < 0.5 \end{cases}, \quad (10)$$

which is a symmetric function around $p = 0.5$.

## 5 RESULTS

Our exploration of hierarchical chains with multi-level abstention (HCMA), based on model-intrinsic uncertainty, yields several findings. First, our nonlinear transformations make Platt scaling much more effective in calibrating LLM output probabilities, yielding a statistically grounded way of performing LLM calibration. Second, mapping out the Pareto frontier of efficient HCMA configurations shows novel risk and efficiency trade-offs that outperform single-model strategies for selective prediction with model-intrinsic uncertainty. Finally, we apply our calibrated correctness prediction methodology to the open-ended QA benchmark TruthfulQA. We discover that chain-of-thought prompting reduces the utility of the abstention signal, highlighting the need for caution in applying established prompting techniques in uncertainty quantification.

### 5.1 TRANSFORMING RAW PROBABILITIES SIGNIFICANTLY ENHANCES PLATT SCALING

Table 1 shows that our modified Platt scaling approach for risk control based on logistic regression strongly outperforms naive Platt scaling. The data shows the results for $n = 50$ training examples on MMLU, showing that our approach is highly data-efficient. Each number is the result of randomly sampling 50 training examples from the MMLU validation set and evaluating on the remaining 1480, repeated 100 times (except in the case of Llama3 405B, where we repeated 500 times).

Table 1: Evaluating logistic regression for correctness prediction using **raw** vs **transformed** token probabilities, on MMLU. Here, precision is the probability that the model *actually* gives the correct answer given that we predict so. In the context of our hierarchical chains with multi-level abstention, high precision is critical for risk control.

| Model | Precision ↑ | | | F1 Score ↑ | | |
|---|---|---|---|---|---|---|
| | Raw | Transformed | %Change | Raw | Transformed | %Change |
| Llama3 1B | 0.671 | **0.798** | 19.00% | 0.651 | **0.701** | 7.62% |
| Llama3 3B | 0.669 | **0.792** | 18.28% | 0.653 | **0.702** | 7.44% |
| Llama3 8B | 0.657 | **0.796** | 21.22% | 0.654 | **0.702** | 7.38% |
| Llama3 70B | 0.663 | **0.793** | 19.61% | 0.656 | **0.704** | 7.29% |
| Llama3 405B | 0.665 | **0.798** | 19.93% | 0.651 | **0.697** | 7.04% |

| Model | Accuracy ↑ | | | Expected Calibration Error ↓ | | |
|---|---|---|---|---|---|---|
| | Raw | Transformed | %Change | Raw | Transformed | %Change |
| Llama3 1B | 0.656 | **0.705** | 7.53% | 0.111 | **0.066** | −40.23% |
| Llama3 3B | 0.651 | **0.700** | 7.53% | 0.117 | **0.072** | −38.58% |
| Llama3 8B | 0.648 | **0.699** | 7.92% | 0.143 | **0.064** | −55.07% |
| Llama3 70B | 0.645 | **0.704** | 9.12% | 0.113 | **0.056** | −50.78% |
| Llama3 405B | 0.653 | **0.702** | 7.55% | 0.064 | **0.053** | −17.56% |

### 5.2 HCMA PROVIDES NOVEL RISK & EFFICIENCY TRADE-OFFS

We study the risk and efficiency trade-offs attainable on MMLU using a HCMA consisting of the Llama3 8B, 70B, and 405B models. To do this, we first tabulate achievable performance metrics – error, cost, and abstention rate – for each 5-dimensional HCMA configuration (involving acceptance thresholds $a_{8B}$, $a_{70B}$ and rejection thresholds $r_{8B}$, $r_{70B}$, and $r_{405B}$) by performing a grid search along the quantiles of the estimated correctness probabilities with 2.5% resolution, resulting in >50 million distinct configurations. For the simulation, we use cost values of 0.3, 0.8, and 5.0 dollars per million tokens, which are close to the current pricing of LLM inference providers (Artificial Analysis (2024)). From the achievable performance profiles, we compute the efficient Pareto frontier using the Skyline operator (Börzsönyi et al. (2001)).

Figure 3 maps out the achievable error rates vs their cost, and colored by the abstention rate. Globally, we observe that selective prediction based on model-intrinsic uncertainty effectively lowers the achievable error rates. At an inference cost of around $1 per million tokens (the cost of Llama3 70B in the model), the graph shows a kink in the slope of the relationship between error and cost, suggesting that increasing usage of Llama3 405B yields diminishing returns in terms of error vs cost.

To evaluate whether the HCMA is useful, it is important to compare performance against baselines. As we are not aware of other approaches incorporating selective prediction into cost-efficient routing, we evaluate against the **single-model strategies** of running selective prediction with the off-the-shelf Llama3 models (8B, 70B, or 405B) using the same method for achieving calibrated correctness prediction. Figure 4 compares the selective prediction performance achieved by the Pareto frontier of the HCMA (dashed lines corresponding to various cost buckets) with those attained by the single-model strategies (solid blue lines).

The plot shows that the HCMA provides two benefits over the single-model strategies. First, the HCMA makes available new error-abstention curves whose price lies between the error-abstention curves attainable with the single-model selective prediction strategies. This means that practitioners have the option of paying "a little bit more" in exchange for better selective prediction performance, without having to jump to the price point of the next biggest model. Second, the HCMA's error-abstention curves in some cases dominate the closest single-model strategy. For example, the error-abstention curve for the $0.6 − $0.9 cost bucket strictly outperforms the Llama3 70B at a similar price. Most strikingly, the HCMA replicates the performance of Llama3 405B, costing $5.0 per million tokens, at an average cost of below $3.0 per million tokens. Alternatively, spending the same amount using the HCMA ($4.7 − $5.0 cost bucket) considerably lowers error for any abstention rate.

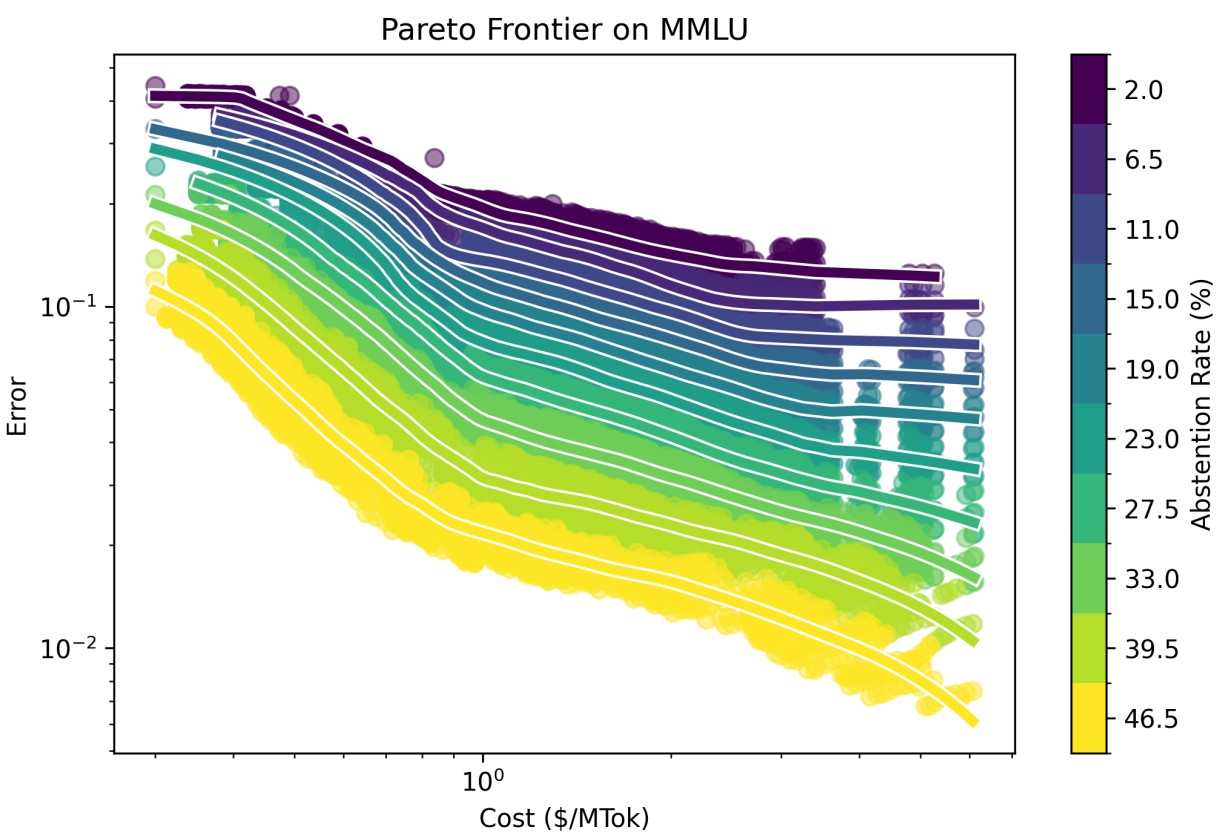

Figure 3: Achievable performance profiles of a HCMA consisting of the Llama3 8B, 70B, and 405B models on MMLU. The graph shows the Pareto frontier of the most efficient HCMA configurations.

### 5.3 EARLY ABSTENTION MAKES LOWEST RISK CHEAPER

To ablate our finding that the HCMA outperforms the selective prediction performances of the off-the-shelf Llama3 models, we consider whether multi-level abstention yields a benefit. Specifically, we compare an HCMA to a constrained version of an HCMA in which only the last model in the chain may abstain, disallowing early abstentions from smaller models.

In the case of a two-model HCMA in which Llama3 8B delegates to 70B ("8B→70B"), we report two findings. First, allowing early abstention can yield a cost advantage in both dollar cost and latency, yielding respective improvements

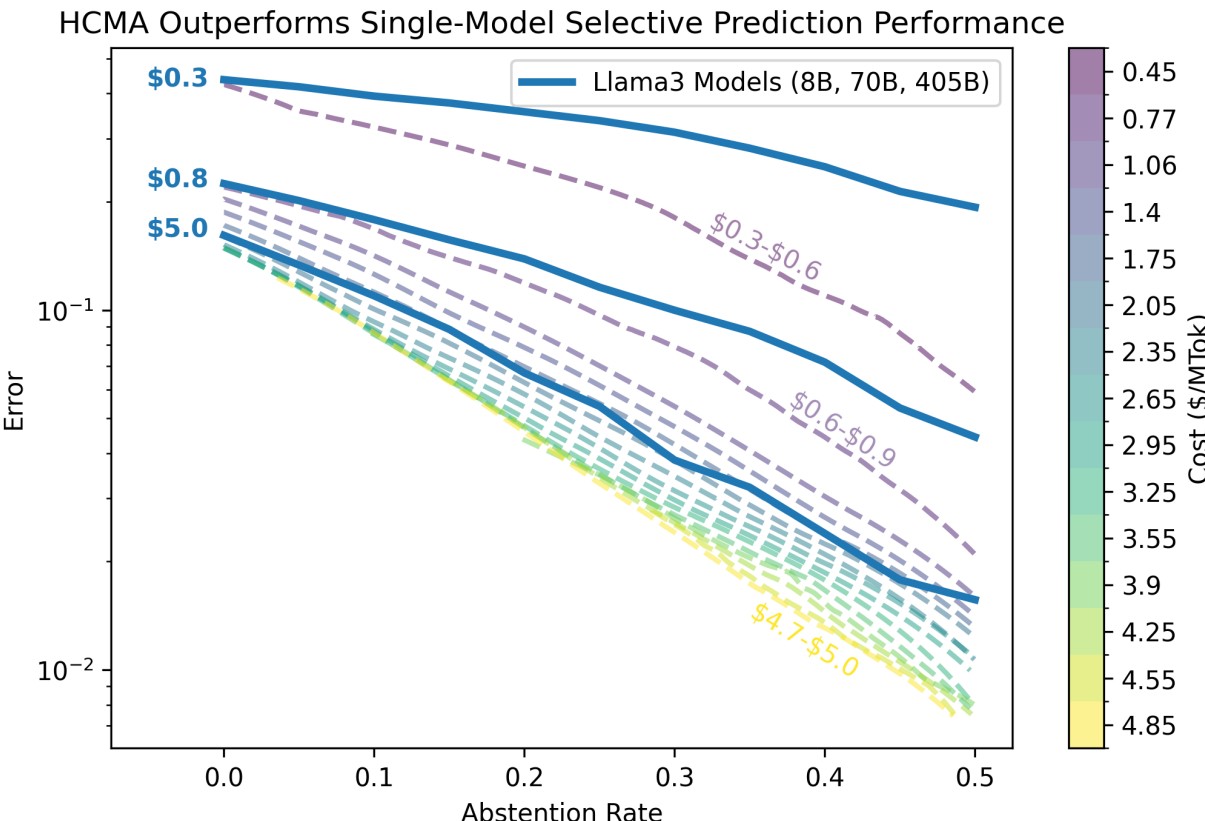

Figure 4: Cost-based view of the Pareto frontier of achievable HCMA performance profiles on MMLU. Each dashed line is the average performance of efficient HCMA configurations in a cost bucket, showing that HCMA outperforms the selective prediction performance of single LLMs (solid blue lines).

of 7% and 6% on MMLU (where we measured latency by querying the Fireworks AI API from our location in California). Second, when imposing a constraint that the average cost must be less than or equal to some value, allowing early abstention strictly dominates the error-abstention curve without multi-level abstention at higher abstention rates between 20% and 50%.

### 5.4 Chain-of-Thought Prompting May Impede Selective Prediction Performance

On TruthfulQA, we observe that **token probabilities derived from chain-of-thought based verification perform poorly with our approach**, since the transformed probabilities form tight clusters around 0 and 1. By contrast, using a zero-shot prompt results in a smooth unimodal distribution, which is well-suited as a confidence signal. See Figure 5.

To ablate this finding, we also tried using a few-shot (but not chain-of-thought) based prompt for verification. The results are intermediate: the distribution of transformed probabilities is unimodal (just as in the zero-shot case) but not quite as symmetric, and the precision/recall curve for predicting incorrect answers is worse. Notably, these results hold **despite the fact that chain-of-thought gives the highest overall accuracy for correctness prediction**. On TruthfulQA, we observed accuracies for correctness prediction of 0.79 for few-shot with CoT, 0.74 for few-shot without CoT, and 0.73 for zero-shot.

These results urge caution in applying established prompting techniques to the task of verifying LLM correctness.

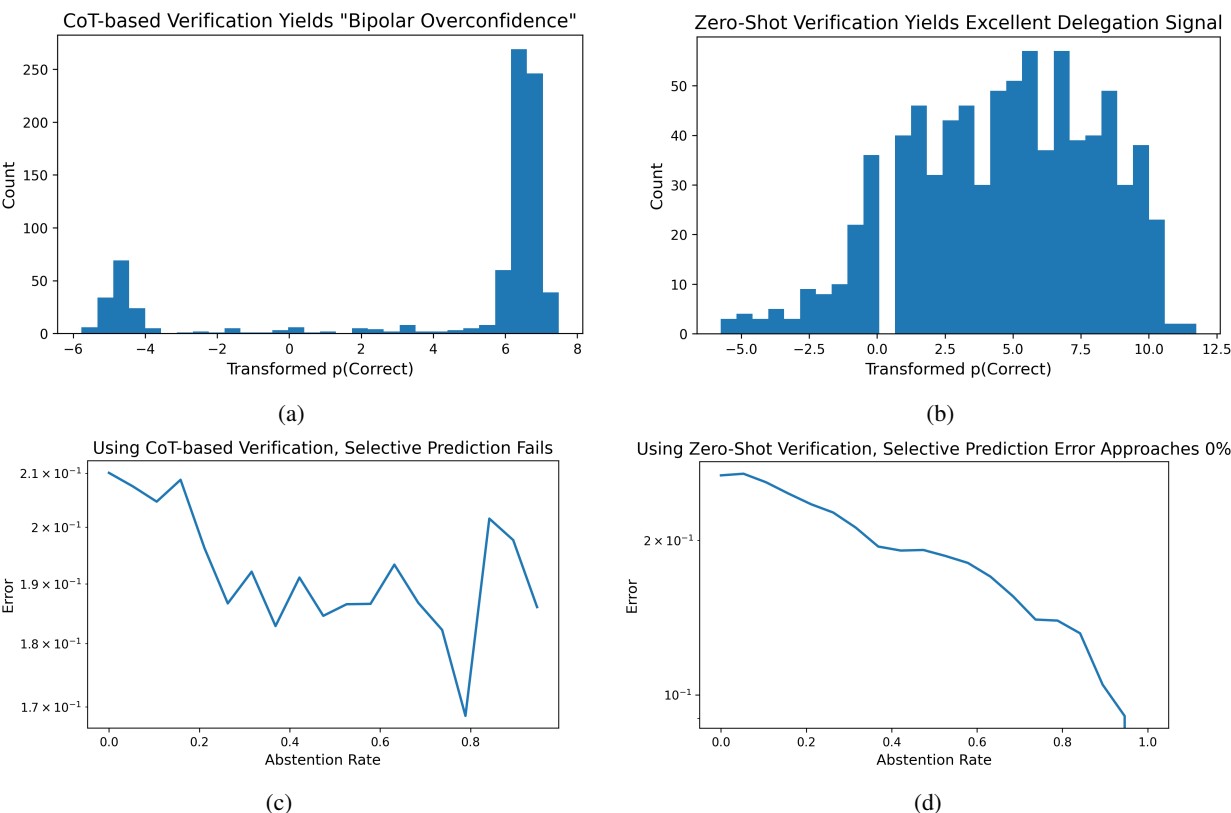

Figure 5: On TruthfulQA, using chain-of-thought prompting for verification leads to highly clustered correctness probabilities (a) that perform poorly as the abstention signal for selective prediction (c). By contrast, using zero-shot prompts leads to a unimodal distribution (b) that performs well as an abstention signal for selective prediction (d).

## 6 Conclusion

Motivated by the observation that differently sized language models directionally agree on the difficulty of queries, we have presented hierarchical chains with multi-level abstention (HCMA), a routed LLM system that incorporates

selective prediction alongside cost efficiency. Given the increased usage of LLMs via black-box API calls, and the efficiency-driven need to avoid repeated LLM sampling, we base our routing and abstention decisions on LLM token probabilities. With our improved Platt scaling approach to calibrating these probabilities, our framework is easily adaptable to new tasks without requiring much labeled data. We show that HCMAs efficiently outperform the selective prediction performance of single LLMs, and that multi-level abstention (as opposed to only abstaining at the end of the chain) is a beneficial part of our system's architecture. Finally, we find that chain-of-thought prompting hinders correctness prediction on TruthfulQA, showing the need for caution in applying established prompting techniques in correctness prediction.

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
