# OpenReview forum: "Efficiently Deploying LLMs with Controlled Risk"
_ICLR.cc/2025/Conference — ICLR 2025 Conference Withdrawn Submission_

### Official Review · Reviewer_31BM · 2024-10-28

**Soundness:** 2
**Presentation:** 1
**Contribution:** 2
**Rating:** 3
**Confidence:** 4

**Summary:**

The paper proposes a cascade of LLMs where each stage can either reject an input, defer it to the next larger LLM, or answer it, depending on their answer token/P(True) probability. The probability is recalibrated using logistic regression and nonlinear transformations. The cascade shows a more favorable error-per-cost tradeoff than using single LLMs on MMLU.

**Strengths:**

* The topic of using model cascades to cut costs is of practical relevance.
* The paper uses cross-validation across 100-500 seeds.

**Weaknesses:**

* The font size is reduced and the margins are made smaller. This is a potential breach of the ICLR guidelines.
* Conceptually, I cannot follow why it is required to recalibrate the LLM token / P(True) probabilities via a logistic regression with a nonlinear transformation of probabilities. All that the method uses in the end are the two threshold values. Since all transformations are monotonic, the thresholds could have also been computed for original probabilities.
* There is no comparison against the routing / abstained prediction SOTA that the paper cites.
* The method works only on one dataset (MMLU) and one cascade of models (Llama 3 models). It fails on TruthfulQA.
* The method requires searching 39^5 (= 90M) hyperparameters. This could be greatly reduced by excluding impossible combinations and using Bayesian optimization.
* Figure 1 has an odd choice of the x-axis values (0, 0.86, 0.982, 0.998, 1.0, 1.0, 1.0, 1.0, 1.0) and does not show any data, just the estimate logistic curves. This makes it hard to tell if the data actually supports the interpretation made from the figure, namely that "differently sized models share a common notion of difficulty"
* Some statements are heavily marketed and oversold. E.g.,
	* "we introduce a nonlinear feature transformation that makes Platt scaling a highly effective calibration technique for LLM token probabilities, providing an alternative to temperature scaling grounded in a rigorous statistical model"
	* "which require only 50 or 100 labeled examples to achieve excellent calibration error (ECE), cutting ECE by 50% compared to naive Platt scaling", I'd advice to remove "excellent" and "only" to make this more objective, since a remaining ECE of 0.05-0.07 is far from excellent. I'd suggest to change 50% to "between 17% and 55%".
* I cannot follow why an arbitrary nonlinear transformation is labeled as statistically grounded ("our nonlinear transformations make Platt scaling much more effective in calibrating LLM output probabilities, yielding a statistically grounded way of performing LLM calibration". Because of the follow-up logistic regression? A logistic regression has no guarantees to produce calibrated values, it just minimizes its loss)
* There is no code released that would allow to replicate the experiments.
* It would be beneficial to report standard deviations in Table 1, since you already used multiple seeds (which is nice!)
* Proposition 1 lacks notation. Is $1_D$ a vector across all decision, with each entry being 1 or 0?

Small notes that did not influence my score and don't need to be rebuttled, I just note them to make the camera-ready better:
* Besides font size and margins, I would suggest to reformat the figures. If you could give them a uniform text size, correct aspect ratio, and potentially use tikz where applicable, removing titles from figures (and put them into the captions). That would improve the presentation a lot.
* Reformat equation 9
* Reformat page 8
* The reference section is misformatted ("Can llms express their un-
certainty? an empirical evaluation of confidence elicitation in llms"). Consider adding double brackets to the .bib entries.
* Typo in line 21: yields drive
* Typo in line 249: hafve
* It's more common in ML literature to have the contributions (§3) be part of the introduction (§1), that would make it easier to digest for a quick reader

**Questions:**

See weaknesses.

---

### Official Review · Reviewer_AoHH · 2024-11-04

**Soundness:** 3
**Presentation:** 3
**Contribution:** 2
**Rating:** 3
**Confidence:** 4

**Summary:**

This paper proposes a new algorithm to route a given query to the model chain comprised of models of the same family but varying sizes such as 1B, 7B, 13B etc. These models are arranged in the increasing order of model size. As inference on larger models is expensive, the query is first routed it to the smallest model in the chain. This model then decides if it should abstain from answering the query altogether on behalf of all the models in the chain, or if it should delegate the query to the next larger model or if it is should answer the query. This routing is determined by the probability of the correctness of that model. If the probability is less than the rejection threshold, then it abstains from answering on behalf of all the entire chain, if it is between the rejection and the acceptance threshold, it delegates the query to the next model in the chain and if it is greater than the acceptance threshold, it answers the query. Thus by using smaller LLMs whenever possible in lieu of using the largest model all the time, they reduce the inference time and the cost for each query.
In order to make sure that the probability computation is calibrated, they employ Platt scaling and adapt it to avoid clustering around probability  value of 1.0.

**Strengths:**

1. The related work is covered in great detail.
2. This paper tries to reduce the cost by utilizing smaller LLMs if they answer the given query correctly rather than always using larger LLMs for each query. They delegate the more difficult queries to larger LLMs or abstain from answering the query altogether if they are not confident enough.

**Weaknesses:**

While the problem they are tackling is quite relevant, the paper lacks sufficient experiments and baselines to demonstrate the efficacy of the proposed method. I have listed a few of my concerns below.
1. How does the modified Platt scaling work in comparison to other uncertainty quantification and probability calibration techniques such as semantic entropy (Kuhn et al.), P_true (Kadavath et al., 2022), Eigen values, Degree, Eccentricity (Lin et al. 2024) and other works listed in the uncertainty quantification part of the related work section. While the authors are performing probability calibration, it is also comparable to uncertainty estimation as the query can be rejected when the uncertainty is high.
2. While you showed the results for various threshold values on one dataset, in a real world scenario, how would one go about in setting the acceptance, rejection threshold values so that it works well for various kinds of queries?
3. In order to demonstrate the generalization, please evaluate the figure 3 and table 1 on more datasets such as helaswag (https://huggingface.co/datasets/Rowan/hellaswag), SQUAD (https://huggingface.co/datasets/rajpurkar/squad) etc and other families of models such as Mistral, Flan-t5, Gemma etc. It is also important to verify if this algorithm works when the differences between model sizes are large or if it would work when the model sizes are opt 350m, opt 1.3b, opt 2.7 b etc.
4. Could you also tabulate the regret wrt accuracy and cost? That is given a model chain comprised of llama 8B, 70B and 405B, we need the ground truth of the smallest possible model that could answer the question and if the model chain should reject the query. Based on this ground truth, you could compute the error and the cost introduced by routing it the wrong model. If the cost is lower than the ground truth, then u can use 0. Only routing it to more expensive models would be penalized. While the plot in Figure 3 demonstrates if the delegated model is correct or not, we are interested in understanding if the algorithm is indeed routing it to the right model.

**Questions:**

1. In general, if more than one family of LLMs exist of various sizes,  such as Mistral 7B, llama 2 7B, Gemma 2b, Opt models listed above as well, how would this work generalize to that setting?
2. As this kind of delegation is more expensive during runtime than training a meta-model to route the query to right model, can you elaborate what are the scenarios where this method could be preferred over that? Perhaps this generalizes better on the domains that are out of the domain of the meta-model’s training data? It would also be nice to bolster your argument with relevant empirical evidence.

---

### Official Review · Reviewer_Pd96 · 2024-11-04

**Soundness:** 1
**Presentation:** 1
**Contribution:** 1
**Rating:** 3
**Confidence:** 2

**Summary:**

The paper introduces Hierarchical Chains with Multi-Level Abstention (HCMA), a framework aimed at improving both efficiency and risk control in deploying LLMs.

**Strengths:**

The proposed HCMA method operates independently of model weights, which allows it to function within API-based LLM query setups.

**Weaknesses:**

### Unclear Motivation
- The motivation for a method that addresses both efficiency and risk control in LLM deployment simultaneously is not clearly explained. It is unclear why existing methods addressing efficiency or risk control separately are insufficient.
- The rationale behind the HCMA approach requires clarification.
- The paper would benefit from a stronger scientific argument that demonstrates a common challenge in efficiency and risk control in LLM deployment, justifying the simultaneous attention to efficiency and risk control of the proposed method.

### Questionable Evaluation
- The paper’s evaluation of “risk control” is primarily based on performance metrics from tasks like MMLU. This choice raises questions about how HCMA’s risk control distinguishes itself from other methods that optimize efficiency through similar performance-cost tradeoffs.
- No baselines from related works are included, limiting the ability to benchmark HCMA’s effectiveness against existing approaches.

### Confusing Presentation
- In Figure 1, the y-axis appears to change despite a fixed x-axis value of 1.0 on the right. The basis for this plot needs further explanation: Is it an extrapolation based on several sample points?
- The text references numerous terms without adequate explanation or citation, such as “abstention rate” (L21), “based on hidden layer embeddings, repeated sampling, and neural-network correctness predictors” (L49), “uncertainty-based delegation” (L92), and "... Platt scaling ... calibration technique" (L93). A clearer introduction to these terms would enhance readability.
- Table 1 lacks a comprehensive analysis of the presented results, leaving the interpretation of findings ambiguous.
- In Figure 3, the x-axis for $/Mtok is presented as a variable, though it is typically a fixed cost for each specific model. This discrepancy requires clarification.

**Questions:**

see the questions mentioned above

---

### Note · Authors · 2024-11-28

**Comment:**

We would like to thank the reviewers for their feedback. We have decided to withdraw this submission.

**Withdrawal Confirmation:**

I have read and agree with the venue's withdrawal policy on behalf of myself and my co-authors.